# Multiple Roles of cAMP in Vertebrate Retina

**DOI:** 10.3390/cells12081157

**Published:** 2023-04-14

**Authors:** Natalia Erofeeva, Darya Meshalkina, Michael Firsov

**Affiliations:** Sechenov Institute of Evolutionary Physiology and Biochemistry, Russian Academy of Sciences, 194223 St. Petersburg, Russia

**Keywords:** cAMP, retina, phototransduction, photoreceptors, ganglion cells

## Abstract

cAMP is a key regulatory molecule that controls many important processes in the retina, including phototransduction, cell development and death, growth of neural processes, intercellular contacts, retinomotor effects, and so forth. The total content of cAMP changes in the retina in a circadian manner following the natural light cycle, but it also shows local and even divergent changes in faster time scales in response to local and transient changes in the light environment. Changes in cAMP might also manifest or cause various pathological processes in virtually all cellular components of the retina. Here we review the current state of knowledge and understanding of the regulatory mechanisms by which cAMP influences the physiological processes that occur in various retinal cells.

## 1. Introduction

In the retina, like in any other tissue, secondary messengers carry out a wide variety of regulatory functions. The central functions of the retina are the reception of light, the conversion of the light signal into an electrical signal, and then the primary processing of visual information before it is transmitted to the brain. Two key cyclic nucleotide monophosphates, cAMP and cGMP, play important but distinct roles in regulating these functions, varying in cell types, effector molecular mechanisms, and even timescales.

There are not many known pathways of cGMP-mediated regulation in the retina, but one of the few is light perception via the phototransduction cascade. In vertebrate photoreceptors, cGMP acts as a regulator of the permeability of plasma membrane channels, the role of which has been well-established, and the pathways of synthesis, hydrolysis, and regulation of these processes have been studied in detail. Changes in cGMP levels occur on the same timescale as the rapid electrical response of the cell. At present, the regulatory role of cGMP is virtually limited to the phototransduction cascade in photoreceptors. In contrast, the regulatory effects of cAMP have been shown for virtually all cellular components of the retina, but the level of mechanistic detail is far from that shown for cGMP in the phototransduction cascade [1,2,3]. Moreover, there is increasing evidence that cAMP may also be involved in the regulation of the phototransduction cascade but on a longer timescale within circadian rhythms. In addition to the phototransduction cascade, the regulatory effects of cAMP have been demonstrated for many other processes in the retina, including circadian cycles, intercellular contacts, retinomotor effects, and so forth. The regulatory role of cAMP in the functioning of various retinal cell types has been the subject of previous reviews [4,5,6]. Here, we continue and extend the description of the regulatory role of cAMP in the retina, focusing on those areas in which there has been the most recent progress.

## 2. cAMP Turnover in the Retina

### 2.1. cAMP Synthesis

cAMP is synthesized from ATP by adenylyl cyclases (ACs) and like in other tissues retinal cAMP signaling is tightly coupled to energy metabolism. The intensity of energy metabolism in the retina is much higher than in the rest of the CNS; for example, it expresses the oxygen transporter neuroglobin 100 times more intensely than the average for brain tissue [7]. Thus, the retina predominantly utilizes oxidative rather than glycolytic pathways due to its higher energy efficiency.

Mammalian ACs are represented by ten isoforms: nine transmembrane ACs(1–9) and one soluble sAC/AC10 [8], which can be divided into four distinct groups according to their characteristic regulatory mechanisms. Several group I isoforms are expressed in photoreceptors and other retinal cells (Table 1). They are activated by the G-protein alpha subunit, G_as_, and Ca^2+^/calmodulin, and inhibited by G_αi/o_ and G_βγ_ [9]. AC1 is abundantly expressed in cone and rod photoreceptors and in retinal ganglion cells. This isoform can also be regulated by phosphorylation at 17 regulatory sites found in high-throughput proteomics studies. Phosphorylation at two sites (S543 and S550 in human AC1) by calcium/calmodulin-dependent protein kinase type IV (CAMK4) expressed by inner retinal neurons has been shown to inhibit the enzyme [10]. The protein kinases C alpha and zeta, which are expressed in abundance in the retina, can directly activate AC1 in an additive manner, although PKC alpha is less potent than PKC zeta [11]. These kinases can also inactivate the G_αi_ signaling pathway by phosphorylation and, thus, indirectly relieve the inhibitory tonus of G_αi_ on AC1 [12]. The post-translational regulation of AC1 expression in the retina requires further investigation, for example, miR351-5p-mediated suppression of AC1 expression, although AC1 contains seed-binding sites for it in 3′-UTR regions, and this miRNA has been shown to increase its expression in retinal degeneration [13]. The presence of AC8 in mouse photoreceptors has been demonstrated by a proteomic study [14]. This enzyme cannot be inhibited by CaM kinase II or IV but can be inactivated by protein kinase A phosphorylation at S115 [15], providing a negative feedback loop in enzyme regulation.

From group II, AC2 is expressed in rod-free knockout mice [16]. In the fish retina, AC2 was found in the inner but not outer segments of photoreceptors [17]. AC7 has been found in photoreceptor cells, but at low levels of expression that change during retinal neovascularization [18]. The presence of AC6 and AC9 has also been demonstrated in photoreceptor cells, but at low expression levels. 

In addition to classical membrane-associated ACs, the human retina also expresses soluble AC (AC10) [19], which functions as a bicarbonate ion sensor [20] and thus reflects tissue metabolic activity. The function of AC10 has been investigated in retinal ganglion cells (RGC), where it provides sufficient levels of cAMP for protein kinase A activation and RGC survival during injury [21]. Photoreceptor survival has also been shown to depend on cAMP levels but the extent to which this is regulated by AC10 remains to be elucidated [22].

### 2.2. Mechanisms of cAMP Degradation by Cyclic Nucleotide Phosphodiesterases

The degradation of cAMP involves the hydrolysis of the phosphodiester bond, resulting in the formation of 5′-AMP [23], which can be further involved in signaling via AMPK [24]. This process terminates cAMP signaling in time and space. In general, cAMP degradation can be mediated by cAMP-specific phosphodiesterases (PDE4, PDE7, and PDE8) and by phosphodiesterases that act on both cAMP and cGMP (PDE1, PDE2, PDE3, PDE10, PDE11) [25]. However, the expression of the latter universal phosphodiesterases is rather low in the human retina. The best-known retinal PDE isoform, PDE6 [26], is strictly specific for cGMP and, therefore, participates in cAMP degradation with very low efficiency [27].

PDE1 has been found in the outer retina [28] and has a higher affinity for cAMP than for cGMP [29] (Table 1). It is activated by calcium-calmodulin. PDE1A is localized to the photoreceptor outer segments, PDE1B is evenly distributed throughout the retina and PDE1C is found in the inner nuclear layer of the retina [29].

PDE4 is expressed in large quantities in the CNS and may therefore play a role in retinal function [30]. Three types of PDE4 are expressed in the inner retina (namely, in bipolar, horizontal, amacrine, and ganglion cells)—PDE4A, PDE4B, and PDE4D—and PDE4B is also found to be abundantly expressed in photoreceptor inner and outer segments [31]. PDE8A and PDE8B are also expressed in the retina but their location remains to be determined (https://www.proteinatlas.org/search/PDE8 (accessed on 12 January 2023)).

## 3. Targets of cAMP Regulation

cAMP regulates an extremely wide range of processes in all cell types in vertebrates, including retinal cells. PKA and exchange protein activated by cAMP (EPAC) are direct effectors of the regulatory effects of cAMP, which affects cellular processes by phosphorylation of a wide range of proteins, or by modulation of guanine nucleotide exchange through activation of small GTPases. The third important target for cAMP regulation is cyclic-nucleotide-gated ion channels [32]. Depending on which proteins are targeted for regulation, cAMP can have rapid or relatively slow effects on cellular processes in the retina. In the latter case, phosphorylation of various transcription factors will lead to changes in the expression of key retinal proteins, such as rhodopsin, transducin, PDE, N-acetyltransferase, and others. PKA consists of an R-subunit dimer (RI α, RI β, RII α, RII β) linked to two C-subunits (Cα, Cβ) [33]. In the inactive state, PKA exists as a tetramer consisting of two regulatory and two catalytic subunits. The catalytic subunit contains an active site that binds and hydrolyses ATP and a binding domain for the regulatory subunit. The regulatory subunit has domains for binding cyclic AMP, a domain that interacts with the catalytic subunit, and an autoinhibitory domain [34]. Each subunit was found to have a different spatial distribution. Cα and Cβ were localized in all cell layers, whereas RIIα and RIIβ were selectively enriched in photoreceptor cells, where both subunits showed a distinct pattern of co-localization with Cα but not with Cβ. Only Cα was observed in photoreceptor outer segments and at the base of the connecting cilium. Cβ, on the other hand, was highly enriched in mitochondria and was particularly prominent in the ellipsoid of cone cells [35]. In addition to the PKA, cAMP utilizes a downstream signal mediated by EPAC. EPAC induces the conversion of GDP-bound inactive Rap into the GTP-bound active form [36,37] The presence of two isoforms of EPAC (EPAC1 and EPAC2) has been shown in horizontal and bipolar cells and for EPAC2 in cone photoreceptors [38], other evidence suggests that EPAC2 is also present in the inner segments of rods [39].

### 3.1. cAMP Impact on Phototransduction Cascade 

In addition to cGMP, a pool of cAMP is also present in the photoreceptors. The reported total amount of cAMP in photoreceptor cells or in rod outer segments varies between 2.2 and 12 pmol/mg protein [40,41,42], which is considerably lower than the cAMP content in the whole rat retina of 138 pmol/mg protein [43]. The total amount of cGMP varies between 42 and 145 pmol/mg [40,41,44,45] which is close to the amount of cGMP reported for the whole rat retina of 44 pmol/mg [43]. Thus, whereas the ratio (total cAMP)/(total cGMP) in the whole retina is about 3, the same ratio in the photoreceptors is 0.06–0.08, or about 1/15 of the total amount of cGMP. The rapid response of photoreceptors to changes in light levels is mediated by the pool of free cGMP, which is about 3% of the total amount of cGMP in the cell [46], meaning that the amount of free cGMP and cAMP are approximately equal.

cGMP is a secondary messenger of phototransduction, and this defines its central role in the operation of the cascade, but the role of cAMP in regulating the phototransduction cascade is still far from being understood. A number of key proteins of the phototransduction cascade, such as the G protein-coupled receptor kinase (GRK) rhodopsin kinase, cGMP phosphodiesterase (PDE6), guanylate cyclase activation protein GCAP, phosducin, and cGMP-regulated channels in the plasma membrane have been shown to carry sites for phosphorylation by protein kinase A (reviewed in [4]).

Vertebrate rhodopsin kinase exists in two functionally distinct forms, GRK1 and GRK7. GRK1 is expressed predominantly in rods but also in some cones, whereas GRK7 is expressed only in cones. PKA-dependent phosphorylation of GRK1 and GRK7 was first shown in vitro [47] and then in vivo for GRK7 in mammalian, amphibian, and fish cones [48], and for GRK1 in mouse photoreceptors [49] (Table 1). PKA-dependent phosphorylation of GRK1 in mouse and GRK7 in Xenopus models was light-dependent [48,49], although cAMP-mediated phosphorylation of GRK1 affects the kinetics of dark adaptation in mouse rod but not cone photoreceptors [50]. Conversely, in zebrafish cones, cAMP-mediated phosphorylation downregulates GRK7 but not its cone-specific ortholog GRK1b [51]. Taken together, these data suggest that at least one key element of the phototransduction cascade, rhodopsin kinase, is downregulated in the dark via inhibition of its activity by PKA-mediated phosphorylation after Ca-mediated rise of cAMP. Thus, the mechanism of phosphorylation is specific for GRK1 in rods and GRK7 in cones and, therefore, may involve different downstream kinases with different specificities [51].

For the other components of the cascade, the mechanisms of regulation of their activity by cAMP have not yet been shown, although they have PKA phosphorylation sites. However, in a frog rod, cAMP levels increased severalfold by forskolin slowing the photoresponse shutoff and significantly increasing the sensitivity of the rod photoresponse [52], which could be explained by an increase in PDE6 activity, a decrease in guanylate cyclase activity, and an increase in Ca^2+^ exchanger current. While the molecular mechanisms that would explain these effects are unknown, elevated cAMP is able to increase the signal-to-noise ratio of photoreceptors by a factor of 2, which might benefit sensitivity during the night phase of the circadian period [53].

Photoreceptor cAMP increases in the dark and decreases in the light phases of the circadian period, and this rhythm is set by the circadian release of dopamine in the inner retina. Dopamine controls cAMP in photoreceptors by regulation of adenylate cyclase 1 expression; thus, currently known mechanisms of cAMP regulation include cAMP synthesis but not cAMP degradation by phosphodiesterases (reviewed in [4,54]. Much less is known about cAMP changes on a faster timescale, comparable to the typical duration of photoresponse development. Sato and colleagues used two-photon live imaging to measure PKA activity in mouse rods in response to relatively short (6 s) and long (10 min) light stimuli in the dark- and light-adapted retina and found a pronounced decrease in PKA activity immediately after light onset, followed by an even greater increase in PKA activity, forming an overshoot with a characteristic time constant of minutes [55]. Sato and colleagues proposed that light onset initiates two opposing mechanisms of cAMP changes—one decreasing cAMP with relatively faster kinetics and one increasing cAMP with much slower kinetics. The first is most likely previously described light-induced, dopamine-mediated negative regulation of cAMP concentration in photoreceptors [54]. The second mechanism that increases cAMP production in response to light onset is still unknown, although the authors have shown that it requires functioning rhodopsin and transducin [55]. On the other hand, the initiation of cAMP production occurs at supersaturating light intensities, suggesting that it is not involved in a late stage of the phototransduction process. Importantly, in contrast to the light-induced suppression of cAMP production, which is thought to be mediated by dopamine produced in amacrine cells, the mechanism of light-induced activation of cAMP production is located in the photoreceptors [55].

In recent years there has been additional evidence that cAMP is actively involved in the regulation of the phototransduction cascade, not only within the diurnal circadian rhythms but also on a much faster time scale, including the development of the photoresponse (summarized in Figure 1). Apart from the rhodopsin kinases GRK1 and GRK7, the other molecular targets of cAMP regulatory effects in the phototransduction cascade remain unknown.

### 3.2. cAMP Changes with a Circadian Period 

Circadian rhythms are significant fluctuations in physiological, biochemical, and other processes with a periodicity of about one day, generated by a genetically encoded molecular clock [56]. The best-known example of circadian rhythms in humans is the sleep-wake cycle [57], the disruption of which can lead to the onset and development of serious metabolic diseases of the body in general and the retina in particular [58]. In mammals, the main pacemaker is the suprachiasmatic nucleus (SCN) of the hypothalamus, for which the main time signal for the molecular clock is light, detected by light-sensitive ganglion cells expressing melanopsin (ipRGCs) [59]. In addition, there are independent peripheral oscillators, such as retinal photoreceptor cells, which have their own circadian machinery. In other words, the retina is able to respond passively to the presence or absence of light and to adapt to different light levels [60]. Even in severe retinal degeneration, the circadian clock is preserved by the reorganization and emergence of dynamic expression of Aa-nat mRNA in the retina [61]. 

Neuromodulators of circadian rhythms in the retina are melatonin and dopamine. In addition, ATP has been proposed as an inter-retinal signal to synchronize the retinal clock [62,63].

Melatonin is synthesized in photoreceptor cells in the dark, from where it diffuses and binds to specific receptors on amacrine, bipolar, horizontal, and ganglion cells [64]. Melatonin leads to the shortening of the rod myoid, lengthening of the cone myoid decreased acetylcholine release, suppression of dopamine synthesis, and adaptation to darkness. Melatonin synthesis depends on a number of factors, including the level of cAMP, which mediates transcriptional and post-transcriptional mechanisms (leading to phosphorylation of CREB, which converges on CRE elements in the Per1 and Per2 promoters) [59]. The level of cAMP increases during the second half of the subjective night and decreases during the second half of the subjective day, which can be explained by the pacemaker activity of AC and the presence of control by the phototransduction cascade. In the retina, melatonin acts mainly by activating two different GPCRs, known as MT1 and MT2, which form functional heterodimers MT1/MT2 that signal by activating the phospholipase C (PLC) protein kinase ζ pathway. The axons of cells expressing Mel1a (MT1) receptors form synaptic contacts with cone terminals that synapse with Mel1b (MT2) receptor-positive OFF-cone bipolar cell dendrites [64]. 

Dopamine is synthesized in amacrine cells when illumination is increased, resulting in effects opposite to those of melatonin [54]. Dopamine also modulates the phototransduction cascade through two independent mechanisms that are mediated by cAMP and calcium, respectively [65]. The G-protein-coupled dopamine receptor D4, which is highly expressed in the rodent and human retina, is under the control of circadian rhythms and plays an important role in dopamine-mediated cAMP degradation, downregulation of PKA activity, and reduction of gap junctional coupling in photoreceptors. Adenosine, which is a nighttime signal with high extracellular concentration at night and low concentration during the day, enhances the phosphorylation of photoreceptor connexin Cx35 through the activation of A2 receptors. A2 receptors induce activation of PKA, and this makes adenosine and dopamine an interdependent reciprocal pair that controls the same pool of cAMP but in opposite directions [66]. It was found that two arrestins are required to modulate the desensitization and internalization of this D4 receptor: a visual arrestin (e.g., ARR4) and a beta-arrestin. Similar processes may occur in cones [67]. Application of dopamine and all agonists (D1R agonist SKF-3839, D2R agonist quinpirole, and D1-D2 receptor heterodimer agonist SKF-83959) to rod outer segments reduces rod sensitivity by slowing the activation rate of the cascade. The D1R agonist has the greatest effect on the photoreceptor response, while the D1-D2 receptor heterodimer agonist modulates intracellular calcium [65]. These data suggest that dopamine exerts its regulatory effect via cAMP-mediated and Ca^2+^-mediated independent mechanisms.

The light-sensitive ganglion cells ipRGCs, which synchronize the circadian clock in the suprachiasmatic nuclei, are also affected by cAMP [68]. Increasing intracellular cAMP levels in ipRGCs increases the duration of the photoresponse and the number of spikes induced by light stimulation. When the level of cAMP in ganglion cells is reduced at night through modulation by adenosine, whose action is mediated by activation of adenosine A1 receptors, there is a reduction in light-induced adhesions. Adenosine A1 receptors are present in retinal ganglion cells as well as in most other retinal layers and exert their action on the AC-cAMP-PKA pathway via G_i₁,₂,₃_, and G_o_-proteins [69]. It has also been shown that this process is part of circadian regulation and may serve to limit the transmission of nocturnal light signals by ipRGCs to the brain [70]. 

### 3.3. The Role of EPAC in Retinal Neuron Survival 

Upregulation of cyclic nucleotide balance is a characteristic feature of degenerative phenomena in the retina [39] and both cGMP and cAMP are subject to changes, although these changes are not necessarily closely related. For example, it has been shown in mouse models that the development of retinal degeneration leads to the complicated interplay between elevated levels of cGMP, cAMP, and EPAC2. In the early stages of pathological development, an increase in cAMP activates EPAC2 and acts as a neuroprotective factor. In parallel, the level of cGMP increases as well, and the increased interaction of cGMP and EPAC2 reduces the activity of the latter, thus decreasing the protection of the photoreceptors [39] (Table 1). This example points to the protective role of EPACs, which has also been demonstrated in many other studies. For example, loss of EPAC1 in the retinal vasculature in EPAC1-KO mice leads to reduced retinal thickness, increased number of degenerated capillaries, and cell loss in all retinal layers [71]. EPAC1, but not EPAC2, regulates inflammatory mediators in retinal endothelial cells [72], and EPAC1 together with PKA can inhibit the purinergic receptor P2X7R-mediated pathway for activation of inflammasome proteins in human retinal endothelial cells [73]. 

EPAC2, but not PKA, has been shown to mediate the regulatory effect of cAMP on presynaptic glycine exocytosis in mouse amacrine cells [74] and the regulation of ascorbate transport from retinal neurons [75]. However, in an animal model of ocular hypertension, activation of EPAC1 was sufficient to induce RGC death [76], again suggesting the ambiguous role of EPAC activation in the development of neurodegenerative processes. 

### 3.4. cAMP in Retinal Neurite Outgrowth

Obviously, an increase in cAMP levels is a systemic stress response of cells to pathological processes caused by trauma or degenerative changes, although during the development of degenerative diseases, it is impossible to say unequivocally whether an increase in cAMP levels is a consequence or a cause of pathological processes. One of the consequences of elevated cAMP levels is increased neurite outgrowth in various cell types, e.g., branching of neurites in salamander rods (but not cones) [77] or growth of rat ganglion cell axons [78]. However, neurite outgrowth itself is not necessarily a positive factor, as it can be associated with the loss of axon guidance in the retina [79,80]. In addition to inherited diseases and trauma, retinal degeneration can be caused by cAMP-induced increased activity of the renin-angiotensin system (RAS) due to hypertension or diabetic retinopathy [81].

The mechanism by which cAMP exerts a protective effect on neurite outgrowth is also poorly understood [78]. Interleukin II [82], stimulation of CLIC1 channel expression [83], stromal cell-derived factor 1 (SDF1), and oncomodulin Ocm [84] have been implicated in this process. High levels of cAMP also promote retinal ganglion cell survival in mice [79,84,85] and rats [78] but lead to photoreceptor death [5] through a mechanism involving neogenin activation [86]. The specificity of cAMP signaling is also shaped by A-kinase anchoring proteins (AKAPs). AKAP anchors PKA and other signaling enzymes to discrete intracellular compartments. In cultured primary RGCs, AKAP is required for survival and axon growth [87], and AKAP signalosomes coordinate cAMP signaling, which enables increased RGC axon growth [85] (Table 1).

cAMP plays a role in photoreceptor development and maturation, which is controlled by a number of interacting signaling pathways. Spontaneous waves of electrical activity are mediated by synaptic transmission between stellate amacrine cells (SACs) and retinal ganglion cells (RGCs), traversing the developing mammalian retina and influencing the structure of central connections [88]. Increasing cAMP levels increases the size, speed, and frequency of these waves. Conversely, inhibition of AC or PKA prevents wave activity [89]. 

Studies of the differential effects of membrane and soluble AC on RGC survival have suggested that soluble AC plays an important role in maintaining cAMP levels and thus in the survival process. Inhibition of soluble AC by KH7 and 2-hydroxyestradiol or its knockdown by siRNA reduces RGC survival and axonal growth. In contrast, inhibition of transmembrane AK by SQ22536, 2,5-dideoxyadenosine, or double knockdown of AC1/AC8 did not significantly reduce RGC survival and axon growth [21].

Modulation of downstream cAMP targets also alters RGC survival. A competitive PKA inhibitor H89 can reduce RGC survival and neurite outgrowth [90], but the same inhibitor did not reduce RGC survival in the presence of forskolin, suggesting that the effect may be PKA-independent [91]. A protective effect against cAMP-induced degeneration has been shown with GM1, a ganglioside that acts by activating mitogen-activated protein kinase (MAPK) and phosphorylating cAMP-responsive element-binding protein (CREB) [92].

**Table 1 cells-12-01157-t001:** The table summarizes the available data on the localization and proven function of the main components of the cAMP-dependent regulation in different compartments of the retina.

	Photoreceptors and Outer Retina	Ganglion Cells and Inner Retina	Retinal Endothelial Cells
PKA	PKA-dependent phosphorylation of GRK1 and GRK7 [47,48,49,50,51,55]	PKA-dependent modulation of ipRGC photoresponse [68,70]. Inhibition of PKA prevents spontaneous wave activity in amacrine retinal ganglion cells [89]	PKA inhibits the activation of inflammasome [73]
EPAC	EPAC1 and EPAC2 (horizontal and bipolar cells [38]), EPAC2 (cone photoreceptors [38], inner segments of rods [39])	EPAC2 mediates a regulatory effect on presynaptic glycine exocytosis in mouse amacrine cells [75]; EPAC1 induces RGC death in a model of ocular hypertension [76]	EPAC1 protects endothelial cells and neurons [39], regulates inflammatory regulators [72], and inhibits activation of inflammasome [73]
AC	AC1, AC2 (mouse cones [16], fish photoreceptors [17]), AC8 (mouse photoreceptors [14], low level of expression AC6,7 and 9	AC1, AC10 (human retina [19])Inhibition of sAC or its knockdown by siRNA reduces RGC survival and axonal growth [21]	
PDE	PDE1 (photoreceptors outer segments [29], outer retina [28]), PDE4 (photoreceptors [31]), PDE6 (photoreceptors [26])	PDE4 (inner retina [31])	
AKAP		AKAP is required for RGC survival and axon growth [87]; AKAP coordinates cAMP signaling and enables RGC axonal growth [85]	

## 4. Conclusions

In all cellular layers of the retina, cAMP acts as a mediator of vital functions at all stages of development, from embryonic to aging. cAMP plays an important, if not central, role in the pathological processes that occur in retinal cells in degenerative processes caused by a variety of causes. A common feature of all these processes is an increase in the intracellular level of cAMP, which in some cases may be a cause and in others a consequence of the pathological processes. The increase in cAMP itself can also act as a protective or pro-degenerative factor, depending on the cell type or the stage of the pathological process. cAMP has a circadian orchestrating and synchronizing effect on a variety of heterogeneous processes that must prepare the retina for the dark or light hours of the day. At the same time, there is evidence that cAMP can have a very local and rapid regulatory effect on an important retinal biochemical complex, the phototransduction cascade in the photoreceptors. 

A detailed understanding of the mechanisms of the regulatory effects of cAMP is of great importance for the development of effective pharmacological agents in ophthalmology.

## Figures and Tables

**Figure 1 cells-12-01157-f001:**
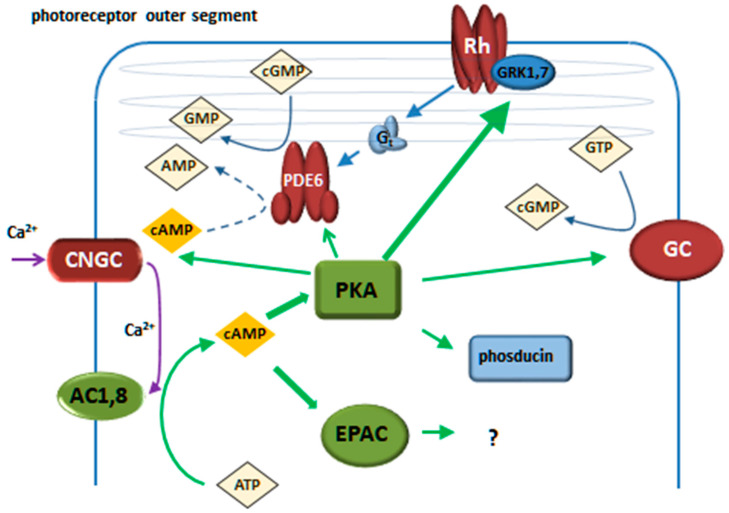
Schematic of the known and putative pathways of cAMP action on the phototransduction cascade. Proteins traditionally implicated in the phototransduction cascade are brown: CNGC—cGMP-gated cyclic nucleotide-gated ion channels, GC—guanylate cyclase, PDE6—cGMP-specific phosphodiesterase type 6, and Rh—rhodopsin; regulatory proteins are blue: G_t_—G-protein transducin, GRK1,7—rhodopsin kinase type 1 or 7, and phosducin. Blue arrows indicate known enzyme activities and interactions between these proteins. Green figures indicate proteins involved in proven or putative loops of cAMP action on the phototransduction cascade: AC(1,8)—adenylate cyclase type 1 or 8, PKA—cAMP-dependent protein kinase, and EPAC—exchange protein directly activated by cAMP. Green arrows indicate proven or putative cAMP action. Hypothetical utilization of cAMP by PDE6 is shown by a dotted line.

## Data Availability

Not applicable.

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
