# Peer review of "Multiple Roles of cAMP in Vertebrate Retina"

_cells, 2023, doi:10.3390/cells12081157_

Round 1

Reviewer 1 Report

The present manuscript interestingly and originally reviews and points out the multiple roles of cAMP in vertebrate retina. Nevertheless, this manuscript overlooks many aspects and does not give a clear pharmacologic message as much as it states possible “divergent changes” in cAMP. Since cAMP may be involved in a longer time scale than cGMP, the manuscript will gain by focusing on the respective and reciprocal participations of both cyclic nucleotides in retina which could not be ignored.

-       Line 35, the canonical refs concerning cGMP implication in cascade vision must be given: Yamazaki, A.,  J. Biol. Chem. 1980; 255, 11619-11624. Pellicone, C, et al. FEBS Letters, 1985 ; 181, 184-188.Ovchinnikov, Y et al., FEBS Letters, 1987; 223, 169-173. Furthermore, it will be of interest to know the molecular organization of rod cGMP Phosphodiesterase 6 that was firstly published (Kameni Tcheudji JF, et al., J Mol Biol, 2001;310, 780-791.

-       Line 85 it is necessary to precise cyclic nucleotide phosphodiesterase (PDE) since, the term “phosphodiesterase” includes many different enzymes as well as snake venom enzyme and consequently it is necessary to specify cyclic nucleotide phosphodiesterases (EC 3.1.4.17) that was defined by  Beavo (1995) in Cyclic nucleotide phosphodiesterases: functional implications of multiple isoforms. Physiol Rev 75, 725–748. 

-       Line 87: It is necessary to add a previous ref introducing  the role of 5’-AMP in AMPK signaling in relationship with cAMP hydrolyzing PDE.

-       Line 119 it is necessary to add the canonical ref characterizing  PKA subunits. 

-        It would be of interest to include in “3. Targets of cAMP regulation” the Cyclic Nucleotide Gated Channel, better than in 3.1. A review on cAMP signaling must be added: J Cell Sci 2001 Jun;114(Pt 11):1971-2. 

-       Lines 126-128: it is necessary to define EPACs and their roles.

-       Ref 49 interestingly pointed out the light induced activation of cAMP production, in that field It will be of interest to study the specific role of cAMP hydrolyzing PDEs such as PDE3 and PDE4, as much as theses PDEs were previously characterized together with PDE6  in chicken pineal gland (Journal of Neurochemistry, 2001, 78, 88-99).

-       Figure 1: according to the manuscript there is no reference concerning cAMP hydrolysis by PDE6, since “it is strictly specific for cGMP” therefore this signaling must be discharged from fig1. Despite giving in the legend the significance of each abbreviation, a commentary is necessary to clarify the figure 1.

-        The figure 2 legend must be developed.

Minor: 

 Line 47:Adenylyl cyclase, instead of adenylate cyclase. Necessity of reviewing and homogenize the printing of various molecular compounds that were printed in minuscules, notably in the bibliography: mRNA instead of mrna, cAMP instead of camp, G protein- instead of g protein-, ……

Unfortunately, it could be not possible to access to ref 1 by using PubMed

 Ref 49 must be completed.

Author Response

  • We thank for the reviewer's helpful advice, which helped us to significantly improve the article. Below are our responses to the reviewer's comments: 

    The present manuscript interestingly and originally reviews and points out the multiple roles of cAMP in vertebrate retina. Nevertheless, this manuscript overlooks many aspects and does not give a clear pharmacologic message as much as it states possible “divergent changes” in cAMP. Since cAMP may be involved in a longer time scale than cGMP, the manuscript will gain by focusing on the respective and reciprocal participations of both cyclic nucleotides in retina which could not be ignored.

    -       Line 35, the canonical refs concerning cGMP implication in cascade vision must be given: Yamazaki, A.,  J. Biol. Chem. 1980; 255, 11619-11624. Pellicone, C, et al. FEBS Letters, 1985 ; 181, 184-188.Ovchinnikov, Y et al., FEBS Letters, 1987; 223, 169-173.

    Done.

     Furthermore, it will be of interest to know the molecular organization of rod cGMP Phosphodiesterase 6 that was firstly published (Kameni Tcheudji JF, et al., J Mol Biol, 2001;310, 780-791.

    Done, inserted

    -       Line 85 it is necessary to precise cyclic nucleotide phosphodiesterase (PDE) since, the term “phosphodiesterase” includes many different enzymes as well as snake venom enzyme and consequently it is necessary to specify cyclic nucleotide phosphodiesterases (EC 3.1.4.17) that was defined by  Beavo (1995) in Cyclic nucleotide phosphodiesterases: functional implications of multiple isoforms. Physiol Rev 75, 725–748. 

    We change the title of section 2.2 , now it is “Mechanisms of cAMP degradation by cyclic nucleotide phosphodiesterases” . The reference to the paper of Beavo is inserted to the next line #87

    -       Line 87: It is necessary to add a previous ref introducing  the role of 5’-AMP in AMPK signaling in relationship with cAMP hydrolyzing PDE.

    See above

    -       Line 119 it is necessary to add the canonical ref characterizing  PKA subunits. 

    PKA subunits are described in line 127 and further.

    -        It would be of interest to include in “3. Targets of cAMP regulation” the Cyclic Nucleotide Gated Channel, better than in 3.1. A review on cAMP signaling must be added: J Cell Sci 2001 Jun;114(Pt 11):1971-2. 

    We add sentence  “The third important target for cAMP regulation are cyclic-nucleotide-gated ion channels [31]” with suggested reference. 

    -       Lines 126-128: it is necessary to define EPACs and their roles.

    Done.  see also reply to Editor

    -       Ref 49 interestingly pointed out the light induced activation of cAMP production, in that field It will be of interest to study the specific role of cAMP hydrolyzing PDEs such as PDE3 and PDE4, as much as theses PDEs were previously characterized together with PDE6  in chicken pineal gland (Journal of Neurochemistry, 2001, 78, 88-99).

    The Reviewer's comment indicates a really interesting potential mechanism of indirect influence on the level of cAMP during a sharp decrease of cGMP level due to its hydrolysis by FDE6. This hypothetical mechanism is described (but only in the sense of secondary decrease) of cAMP level due to activation of PDE3 during removal of inhibitory influence of cGMP. However, in the case of photoreceptor this scheme is too hypothetical, firstly, because it does not explain the growth of cAMP, and besides, the presence of PDE3 in the outer segments of photoreceptors is not shown.

    -       Figure 1: according to the manuscript there is no reference concerning cAMP hydrolysis by PDE6, since “it is strictly specific for cGMP” therefore this signaling must be discharged from fig1. Despite giving in the legend the significance of each abbreviation, a commentary is necessary to clarify the figure 1.

    The mechanism of cAMP hydrolysis by PDE6 is hypothetical, but possible in principle. We do report in the text (line 102, referred to MS version downloaded after review) that PDE6 is highly specific for cGMP, but we also mention  that FDE6 can hydrolyze cAMP with low specificity. In Figure 1, we have replaced the arrow indicating hydrolysis of cAMP by PDE6 with a dotted line and added a corresponding explanation in the caption to Figure 1. 

    -        The figure 2 legend must be developed.

    We substitute Fig.2 with a Table 1 .

    Minor: 

     Line 47:Adenylyl cyclase, instead of adenylate cyclase. 

    Done.

    Necessity of reviewing and homogenize the printing of various molecular compounds that were printed in minuscules, notably in the bibliography: mRNA instead of mrna, cAMP instead of camp, G protein- instead of g protein-, ……

    Done

    Unfortunately, it could be not possible to access to ref 1 by using PubMed

     Ref 49 must be completed.

    Done

Reviewer 2 Report

In the present manuscript the authors review the literature on the cAMP cascade in the vertebrate retina. The authors discuss recent and less recent literature and make an effort to provide a view of the state of the art. 

- In the description of the cAMP signalling cascade the authors focus on synthesis and degradation of the messenger itself. It would be more complete if they could discuss other determinants of the cAMP actions such as AKAPs (compartmentalisation) and Phosphatases (termination).

- It would be also helpful to integrate the written parts on the cAMP components in the retina with a table (for example summarising the component, its function and the study that identified it).

English: Moderate editing is needed before final publication. The manuscript results legible for the most parts, however long sentences and sometimes the improper use of terms (for example the authors refer to the second messenger cascades as "chains") can make the reading difficult. 

Figures 2: It is of difficult interpretation- The authors should consider to modify it or substituted with a table. 

Author Response

Thank you very much ! below are our replies to the comments and suggestions

In the present manuscript the authors review the literature on the cAMP cascade in the vertebrate retina. The authors discuss recent and less recent literature and make an effort to provide a view of the state of the art. 

In the description of the cAMP signalling cascade the authors focus on synthesis and degradation of the messenger itself. It would be more complete if they could discuss other determinants of the cAMP actions such as AKAPs (compartmentalisation) and Phosphatases (termination).

There is not much known yet on the role of AKAPs in the retina, but we added a data on neuronal growth

- It would be also helpful to integrate the written parts on the cAMP components in the retina with a table (for example summarising the component, its function and the study that identified it).

The Table 1 which summarizes the available data on the localization and proven function of the main components of the cAMP-dependent regulation is added.

English: Moderate editing is needed before final publication. The manuscript results legible for the most parts, however long sentences and sometimes the improper use of terms (for example the authors refer to the second messenger cascades as "chains") can make the reading difficult. 

Done, we checked and reviewed the article again

Figures 2: It is of difficult interpretation- The authors should consider to modify it or substituted with a table.

We substitute Fig.2 with a Table 1.